# Ultrafast self-trapping of photoexcited carriers sets the upper limit on antimony trisulfide photovoltaic devices

Zhaoliang Yang[1,6], Xiaomin Wang[2,6], Yuzhong Chen[1,6], Zhenfa Zheng [3], Zeng Chen[1], Wenqi Xu[4], Weimin Liu[4], Yang (Michael) Yang [5], Jin Zhao [3], Tao Chen[2]* & Haiming Zhu [1,5]*

Antimony trisulfide ($Sb_2S_3$) is considered to be a promising photovoltaic material; however, the performance is yet to be satisfactory. Poor power conversion efficiency and large open circuit voltage loss have been usually ascribed to interface and bulk extrinsic defects By performing a spectroscopy study on $Sb_2S_3$ polycrystalline films and single crystal, we show commonly existed characteristics including redshifted photoluminescence with 0.6 eV Stokes shift, and a few picosecond carrier trapping without saturation at carrier density as high as approximately $10^{20}$ cm$^{-3}$. These features, together with polarized trap emission from $Sb_2S_3$ single crystal, strongly suggest that photoexcited carriers in $Sb_2S_3$ are intrinsically self-trapped by lattice deformation, instead of by extrinsic defects. The proposed self-trapping explains spectroscopic results and rationalizes the large open circuit voltage loss and near-unity carrier collection efficiency in $Sb_2S_3$ thin film solar cells. Self-trapping sets the upper limit on maximum open circuit voltage (approximately 0.8 V) and thus power conversion efficiency (approximately 16 %) for $Sb_2S_3$ solar cells.

[1] Centre for Chemistry of High-Performance & Novel Materials, Department of Chemistry, Zhejiang University, Hangzhou, Zhejiang 310027, China. [2] CAS Key Laboratory of Materials for Energy Conversion, Department of Materials Science and Engineering, University of Science and Technology of China, Hefei, Anhui 230026, China. [3] Department of Physics, University of Science and Technology of China, Hefei, Anhui 230026, China. [4] School of Physical Science and Technology, ShanghaiTech University, Shanghai 201210, China. [5] State Key Laboratory of Modern Optical Instrumentation, College of Optical Science and Engineering, Zhejiang University, Hangzhou 310027 Zhejiang, China. [6] These authors contributed equally: Zhaoliang Yang, Xiaomin Wang, Yuzhong Chen. *email: tchenmse@ustc.edu.cn; hmzhu@zju.edu.cn

The exploration of semiconductor material for low-cost, stable, and efficient thin-film photovoltaics has been a key target for solar energy conversion. Among them, Cu(In, Ga)Se$_2$, CdTe, and organic–inorganic hybrid perovskites play the leading role with power-conversion efficiencies (PCEs) above 20%, but on the other hand show various limitations imposed by element scarcity, stability, and environmental concerns. Recently, binary semiconductors antimony chalcogenides including Sb$_2$S$_3$ and Sb$_2$Se$_3$ emerge as promising materials due to their ideal bandgaps ($E_g$ of 1.7 eV for Sb$_2$S$_3$ and 1.2 eV for Sb$_2$Se$_3$), large absorption cross-section, earth-abundance and environmental-friendly and stable characters[1–3]. They have fixed orthorhombic phase with infinite one-dimensional (1D) ribbons along the [001] (or c) direction (Fig. 1a for Sb$_2$S$_3$), avoiding complicated phase control during processing. In particular, Sb$_2$S$_3$ with $E_g$ of 1.7 eV has been considered as a perfect component for the top subcell in Si-based tandem solar cells[2].

Sb$_2$S$_3$ and Sb$_2$Se$_3$ solar cells of both sensitized and planar configurations have been extensively reported in recent years[4–16]. However, their solar cell performances remain unsatisfactory to date[2,3]. The record PCE is 7.5%[5] and 9.2%[9] for Sb$_2$S$_3$ and Sb$_2$Se$_3$ solar cells, respectively, much lower than that of CdTe or perovskite thin-film solar cells with similar bandgaps. While fill factor (FF) up to 70% and near-unity internal quantum efficiency have been achieved, the open-circuit voltage ($V_{oc}$) is surprisingly low, regardless of fabrication and pre/post-treatment methods[1–3]. For example, using either chemical bath deposition, thermal evaporation, or atomic layer deposition, $V_{oc}$ for Sb$_2$S$_3$ solar cell always falls into between 0.6 and 0.8 V with a record value of 0.77 V[17], which is only half of theoretical thermodynamic limit (1.4 V for $E_g$ equals to 1.7 eV) under AM1.5 irradiance[2].

The large $V_{oc}$ loss in Sb$_2$S$_3$ solar cell has been generally attributed to the surface/interface trap states and/or defects/impurities in bulk[18–22]. Various extrinsic defects have been invoked, including surface sulfide state, interfacial states at Sb$_2$S$_3$-electrode contact, antimony oxides formation, and sulfur vacancy[18–22]. These defect states are speculated to trap carries and accelerate their recombination, leading to low density of photoexcited carriers in semiconductors thus low $V_{oc}$[2,18,20]. Previous transient absorption (TA) and time-resolved terahertz spectroscopy measurements on Sb$_2$S$_3$ (Sb$_2$Se$_3$) nanocrystals and polycrystalline films indeed observed picosecond carrier localization/trapping process[19,22,23]. In the face of nearly clamped $V_{oc}$ loss after significant efforts on material optimization and device engineering, a critical question naturally arises on whether there is any intrinsic limitation on this semiconductor as photovoltaic material.

Here, we seek to answer this critical fundamental question based on spectroscopic study of excited-state carrier properties in Sb$_2$S$_3$. By comparing polycrystalline Sb$_2$S$_3$ film of three different growth methods and high-quality stoichiometric Sb$_2$S$_3$ single crystals, we observe strongly Stokes-shifted PL and ultrafast picosecond carrier-trapping process in all samples. Saturation of trapped carrier is not observed at carrier density as high as $10^{20}$ cm$^3$, which is too large to be related to extrinsic impurities or defects. Together with polarized trap emission from single crystals, these results strongly suggest that photoexcited carriers are self-trapped by lattice distortion in Sb$_2$S$_3$. This intrinsic self-trapping explains well the 0.6 V $V_{oc}$ loss and thermally activated carrier transport and ultimately sets the upper bound for the PCE in Sb$_2$S$_3$ and Sb$_2$Se$_3$ photovoltaic devices.

## Results

**Optical study of polycrystalline films.** We first prepared Sb$_2$S$_3$ polycrystalline thin films via in situ hydrothermal growth on CdS/FTO substrate with CdS as electron transport layer in solar cell (see the Methods section for details)[10]. This method has been employed to grow high-quality Sb$_2$S$_3$ films for solar cells, and the thickness of Sb$_2$S$_3$ layer can be controlled by reaction time and temperature[10,24]. The atomic force microscopy (AFM) image

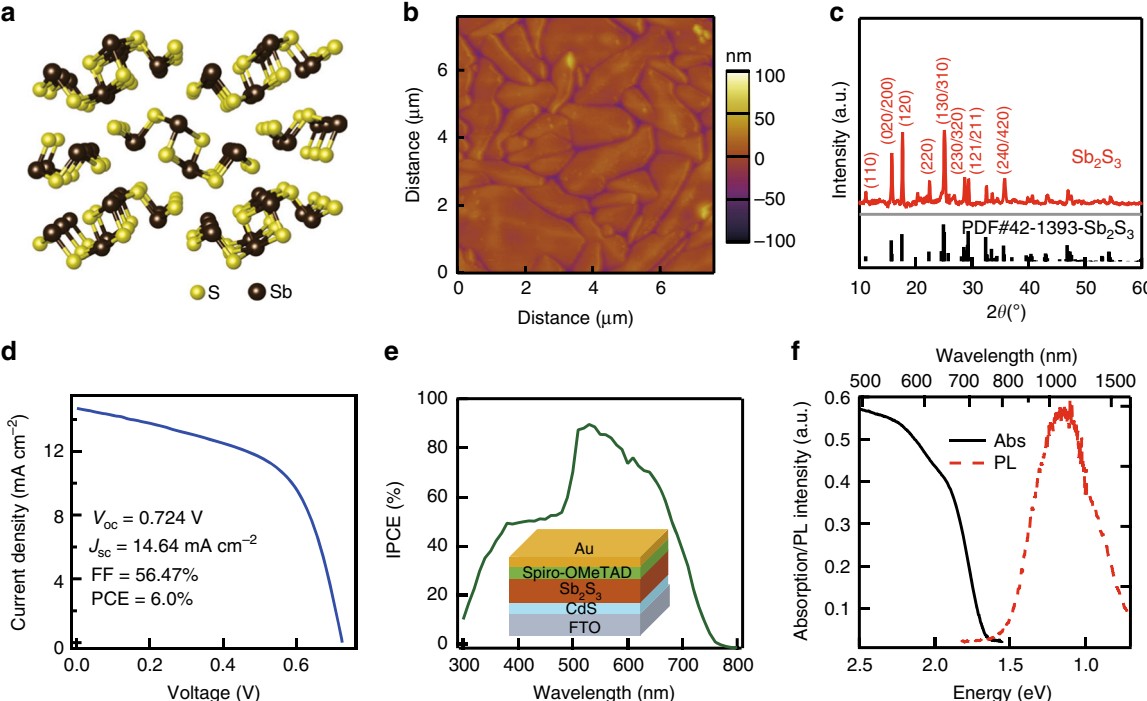

**Fig. 1** Characterization of hydrothermal grown Sb$_2$S$_3$ polycrystalline film. **a** Perspective review of Sb$_2$S$_3$ crystal structure projected on the [001] (or ab) plane. **b** AFM image and (**c**) XRD pattern of hydrothermal Sb$_2$S$_3$ polycrystalline film on CdS/FTO substrate. **d** J–V curve and (**e**) IPCE curve of a representative Sb$_2$S$_3$ solar cell. Inset: FTO/CdS/Sb$_2$S$_3$/Spiro-OMeTAD/Au solar cell device structure. **f** Absorption and PL spectra of hydrothermal Sb$_2$S$_3$ thin film

shown in Fig. 1b indicates that as-grown $Sb_2S_3$ film is smooth and compact with large grain size. The crystal structure of $Sb_2S_3$ film was confirmed by X-ray diffraction (XRD) (Fig. 1c), and diffraction peaks at 15.7°, 17.6°, 25.0°, and 28.6° can be indexed to orthorhombic stibnite structure (JCPDS #42-1393). The photovoltaic properties of hydrothermal grown $Sb_2S_3$ film (thickness of ~300 nm) was examined by assembling FTO/CdS/$Sb_2S_3$/Spiro-OMeTAD/Au structured solar cell device (Fig. 1e inset). The J–V curve and incident photon-to-electron conversion efficiency (IPCE) curve of $Sb_2S_3$ thin-film solar cell under AM1.5 illumination are shown in Fig. 1d and e, respectively. The solar cell exhibits a $V_{oc}$ of 0.72 V, $J_{sc}$ of 14.6 mA cm$^{-2}$, FF of 56.5%, and PCE of 6%, among the top values for planar $Sb_2S_3$ thin-film solar cell[2]. IPCE curve shows a peak value >80% at 510–580 nm, and drops in shorter wavelength due to CdS buffer layer and longer wavelength due to reduced absorption. All these characterizations indicate high-quality $Sb_2S_3$ thin film by hydrothermal growth for photovoltaic devices.

The absorption and photoluminescence (PL) spectra of $Sb_2S_3$ thin-film (thickness of 160 nm) are shown in Fig. 1f. The absorption spectrum exhibits an onset at ~1.65 eV (750 nm) and a weak absorption peak at ~1.9 eV (650 nm). The absorption onset agrees well with the onset of IPCE curve, corresponding to the bandgap of the $Sb_2S_3$ film[2,25]. According to literatures and our calculation (Supplementary Fig. 1), $Sb_2S_3$ is an indirect bandgap semiconductor with indirect gap of 1.7 eV (the precise value depends on sample conditions) and direct gap only slightly higher (80 meV)[2]. PL property of $Sb_2S_3$ has been barely reported. Strikingly, we observed a strongly red-shifted and broad PL peak at 1.15 eV under CW excitation (532 nm), albeit with weak intensity. This corresponds to a Stokes shift as large as ~500 meV, which can only be ascribed to emission from trap states rather than band-edge states.

To directly probe photoexcited carrier dynamics in a $Sb_2S_3$ polycrystalline film, we performed time-resolved TA measurements[19,25]. We excited and created carriers in $Sb_2S_3$ thin film with 2.1 eV pump pulse and after a certain delay time, probed it with either visible white light continuum or mid-IR probe pulse. While the visible photon mostly probes the interband electronic transitions, low-energy mid-IR photon (e.g., 5 μm) is dominated by Drude response of free carriers (Supplementary Fig. 2)[26]. Combining visible and mid-IR probe can provide a comprehensive picture about carrier dynamics in semiconductors. We note both free electron and hole contribute to TA signal, and their relative contribution is inversely proportional to their effective masses[26]. The 2D color plot of TA spectra of the $Sb_2S_3$ polycrystalline film is shown in Fig. 2a. We observed clear spectral evolution with increasing delay time, indicating the presence of multiple transient absorbing species and the conversion between them after photoexcitation. For TA results with spectral and temporal separated species, it is common to do analysis with sets of discrete time and wavelength. For a complex TA result, singular value decomposition (SVD) based on time–wavelength separability provides a facial and general method to describe TA result with the minimum number of transient species (base spectra) on a completely model-free basis[27]. The emergence and evolution of the species can be followed individually with time (base time traces). In order to disentangle transient species and characterize the photoexcitation dynamics in $Sb_2S_3$, we analyzed the TA data by the SVD method.

The TA data of $Sb_2S_3$ polycrystalline film can be well described by two principle components (A and B) with spectra and associated kinetics shown in Fig. 2b, c, respectively. The component A is dominated by bleach of optical transition with peak at 650 nm, which forms instantaneously after photoexcitation and decays in ps. The decay of component A closely follows

the decay of mid-IR kinetics which directly probes free carrier population in the $Sb_2S_3$ film. Therefore, the component A can be assigned to photoexcited free carriers in $Sb_2S_3$ film which bleaches the ground-state transitions through both band filling and Columbic effect (e.g., screening, band renormalization)[26]. The other component B is dominated by induced absorption band with peaks at 700 nm and 565 nm. Interestingly, component B rises gradually in ps and its rising coincides with decay of A component/mid-IR kinetics, indicating that the decay of free carrier at band-edge leads to new transient species in the $Sb_2S_3$ polycrystalline film. Together with the trap emission, component B can be safely assigned to trapped carrier which provides a direct measure for trapped carrier population. Similar induced absorption feature has also been observed in photoexcited $Sb_2S_3$ nanocrystalline film and ascribed to sulfide radical (S•)[19]. Based on these steady-state and time-resolved optical measurements, we can conclude that photoexcited free carriers (including both electrons and holes) in the $Sb_2S_3$ polycrystalline film get trapped in tens of ps, leading to trapped carrier-induced absorption in visible and strongly red-shifted near-IR PL. The detailed carrier-trapping kinetics will be analyzed later with a rate equation model.

To gain more hints about the nature of the trap states discussed above, we varied the excitation fluence thus the transient carrier density over a large range (1.5 × 10$^{18}$ cm$^{-3}$ to 8 × 10$^{19}$ cm$^{-3}$). If the trapping process in the $Sb_2S_3$ polycrystalline film are due to extrinsic defects, e.g., surface states, impurities, or atomic vacancies, the saturation of trapped carrier-induced absorption (component B) would be expected when the trap states are filled. We plotted the maximum amplitude of B component as a function of photoexcitation carrier density (Fig. 2d). Interestingly, we did not observe any saturation signature even at carrier density approaching 10$^{20}$ cm$^{-3}$. Such large trap density suggests carrier trapping in $Sb_2S_3$ is likely intrinsic.

**Optical study of single crystal**. To minimize possible deterioration by extrinsic defects and reveal the intrinsic photoexcited carrier properties in $Sb_2S_3$, we turn to zone-refined stoichiometric $Sb_2S_3$ single crystals grown by chemical vapor transport[28]. As shown in Fig. 3b inset, $Sb_2S_3$ single crystals have needle-like shape with length of a few cm (along c-axis direction) and width/height of ~1 mm, consistent with its quasi-1D crystal structure. Transmission X-Ray Laue photograph of $Sb_2S_3$ single crystal indicates high crystalline quality. We characterized the trap density of $Sb_2S_3$ single crystal by space charge-limited current method, which shows a very low trap density of 6.8 × 10$^9$ cm$^{-3}$ (Supplementary Fig. 3 and Supplementary Note 1). To probe the sample with transmitted light, we exfoliated the single crystal to an optically thin flake with thickness of 130 nm (Supplementary Fig. 4) for both steady state and TA measurements. The anisotropic crystal structure of $Sb_2S_3$ allows perfect cleavage perpendicular to the b-axis[2].

The absorption spectrum of $Sb_2S_3$ single-crystal flake (Fig. 3a) exhibits an onset at 1.7 eV (730 nm) and a peak at 1.9 eV (650 nm), similar to that of the polycrystalline film. The PL of $Sb_2S_3$ single crystal under 532 nm CW excitation is also strongly red-shifted and broad with a peak at 1.1 eV, which represents substantial energy loss (0.6 eV) and can be assigned to trap emission. An important clue about trap state comes from examining trap emission polarization as a function of angle (θ) relative to the crystal c-axis (Fig. 3b). Surprisingly, the trap emission exhibits strong polarization along the c-axis, which can be fit by cos$^2$θ function with a constant offset. The degree of polarization can be calculated by $(I_\parallel - I_\perp)/(I_\parallel + I_\perp)$ to be 24%. This indicates transition associated with trapped hole and

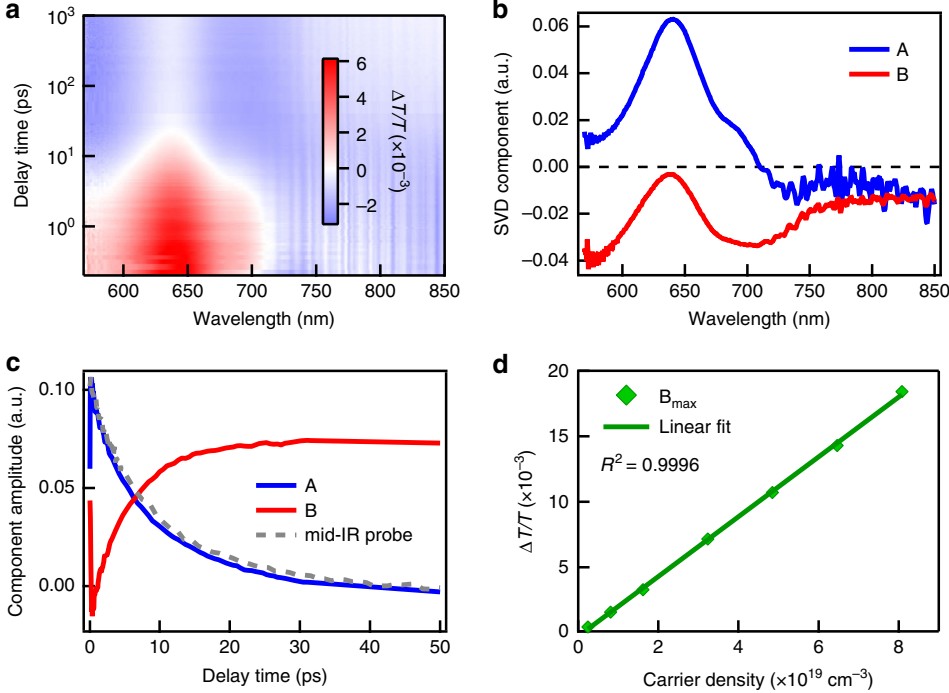

**Fig. 2** TA study of hydrothermal grown the $Sb_2S_3$ polycrystalline film. **a** 2D color plot of TA spectra of as-grown $Sb_2S_3$ polycrystalline film. **b** Principle spectral components and (**c**) associated kinetics from SVD analysis. Also shown in Fig. 2c is the mid-IR (5 μm) probe kinetics (gray-dashed line). **d** Maximum TA signal of B component (trapped carrier induced absorption) as a function of photoexcited carrier density and its linear fitting with $R^2$ equal to 0.9996

electron are preferentially orientated along the *c*-axis in $Sb_2S_3$ single crystal. Similar polarized near-IR PL was also observed from exfoliated $Sb_2S_3$ single crystal thin flakes, precluding the geometry effect as polarization origin. This polarized trap emission is unlikely due to surface states or impurities which usually would be randomly distributed and lead to isotropic trap emission.

More information about trap nature is provided by TA measurement on $Sb_2S_3$ single-crystal flake. The 2D plot of TA spectra is shown in Fig. 3c, which was analyzed by SVD method as above. The principle component spectra and associated kinetics are shown in Fig. 3d, e. Similar to the polycrystalline film, SVD analysis on $Sb_2S_3$ single-crystal flake yields a component A corresponding to photoexcited free carriers and a component B which is mainly contributed by induced absorption of trapped carriers. The main spectral features of principle components are similar in the $Sb_2S_3$ polycrystalline film and single-crystal flake with small variation likely due to inhomogeneous polycrystalline nature in former, justifying SVD method and suggesting their similar spectral origin. As shown in 3e, the decay of free carrier was confirmed with mid-IR probe and its decay process is also accompanied by the rise of trapped carrier-induced absorption. We varied the photoexcited carrier density ($2 \times 10^{18}$ cm$^{-3}$ to $1.2 \times 10^{20}$ cm$^{-3}$) for $Sb_2S_3$ single-crystal flake and did not observe any saturation of trapped carrier induced absorption even when the carrier density was above $10^{20}$ cm$^{-3}$ (Fig. 3f), which is too large to be related to extrinsic defects for $Sb_2S_3$ single crystal with a defect density of $6.8 \times 10^9$ cm$^{-3}$.

We performed similar steady state and transient optical measurements on the spin-coated and thermal-evaporated $Sb_2S_3$ polycrystalline films (see Supplementary Fig. 5 and Supplementary Fig. 6) and also observed a near-IR PL with ~0.6 eV Stokes shift and ultrafast carrier trapping to form similar induced absorption without any saturation. The close resemblance between $Sb_2S_3$ polycrystalline films of three different growth

methods and $Sb_2S_3$ single crystal on carrier trapping properties, together with polarized trap emission in $Sb_2S_3$ single crystal, strongly suggest that ultrafast carrier trapping in $Sb_2S_3$ is associated with intrinsic self-trapping, instead of extrinsic defects as usually invoked.

## Discussion

As was first introduced by Landau[29], in a material with soft lattice and strong carrier–phonon coupling, free carriers (electrons or holes), or excitons (bound electron–hole pairs) can be trapped within potential wells produced by local lattice distortion, forming a quasiparticle called "polaron"[30]. When the short-range deformation–potential interaction is dominant over long range interaction, a small polaron forms as self-trapped carrier or exciton is localized within unit cell. Self-trapping as small polaron could occur even in a perfect crystal and create a transient defect state in bandgap by lattice deformation, leading to substantial energy loss and a Stokes-shifted PL (Fig. 4c).

Self-trapping is favored in materials with strong carrier–phonon interaction and small elastic constant, and has been observed in metal halide (e.g., NaCl), oxide (e.g., $SiO_2$) and chalcogenide crystals (e.g., $As_2Se_3$)[31,32]. The elastic constant of $Sb_2S_3$ has been calculated[33], and the value (~40) is as small as that of $SiO_2$ and NaCl crystals[32]. The Huang-Rhys parameter of $Sb_2S_3$, which reflects carrier–phonon interaction, was estimated to be 38.5, which is as large as that of NaCl and $Cs_2AgInCl_6$[34] where STE has been demonstrated (Supplementary Note 2). Dimensionality also plays an important role in self-trapping. Compared with three-dimensional system, free carrier in (quasi) 1D system has been predicted to be intrinsically unstable and tends to be self-trapped without barrier[31]. As it happens, $Sb_2S_3$ has a quasi-1D crystal structure composed with $(S_4S_6)_n$ ribbons stacked with van der Waals interaction. Therefore, carrier/exciton self-trapping is very likely in $Sb_2S_3$ and can explain all the optical

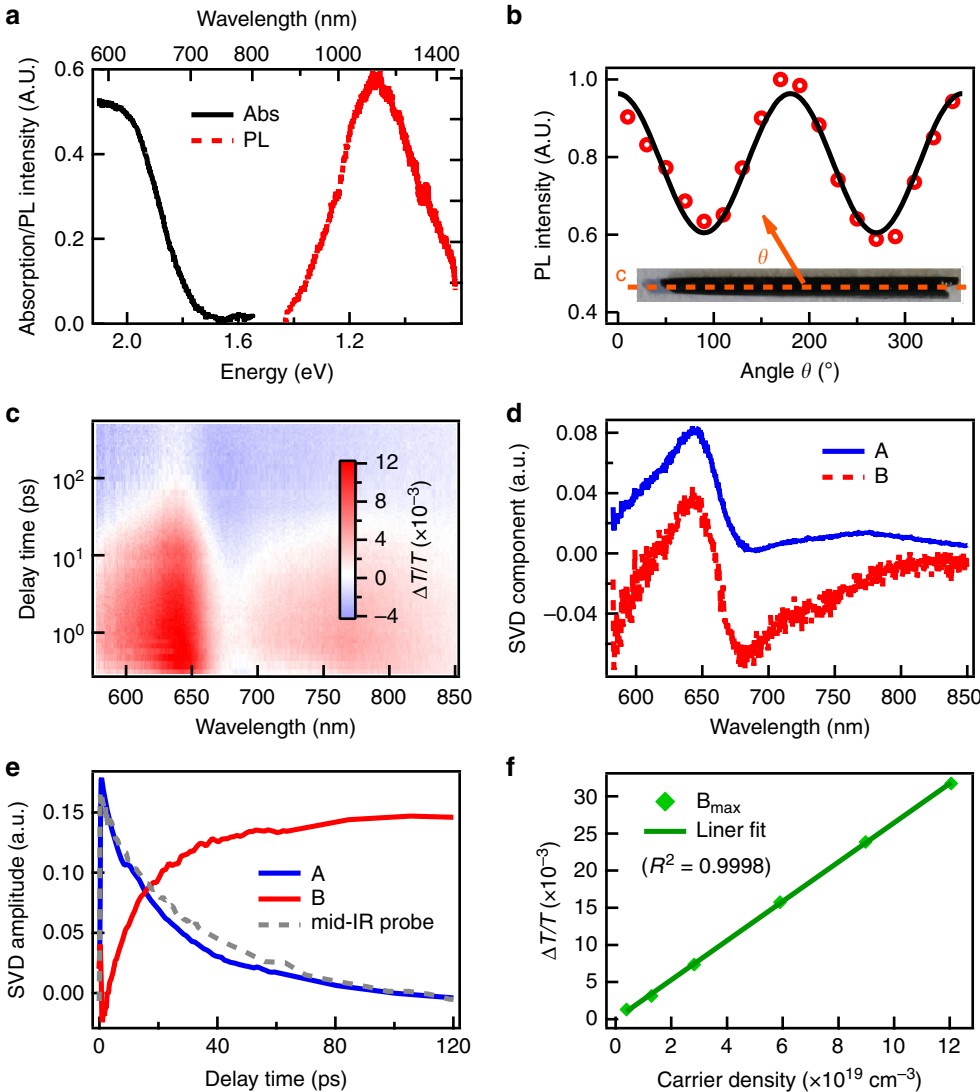

**Fig. 3** Steady state and TA study of Sb$_2$S$_3$ single crystal. **a** Absorption and PL spectra of Sb$_2$S$_3$ single crystal. **b** PL intensity from a Sb$_2$S$_3$ single crystal as a function of polarization detection angle ($\theta$) and fitting with cos$^2\theta$. $\theta$ is defined to be 0 when detection polarization is along the crystal c-axis direction. Inset: optical image of a needle-like Sb$_2$S$_3$ single crystal. **c** 2D color plot of TA spectra of Sb$_2$S$_3$ single crystal. **d** Principle spectral components and (**e**) associated kinetics from SVD analysis. Also shown in Fig. 3e is the mid-IR (5 μm) probe kinetics (gray-dashed line). **f** Maximum TA signal of B component as a function of photoexcited carrier density and its linear fitting with $R^2$ equal to 0.9998

properties and photoexcitation dynamics we observed, including strongly red-shifted trap emission, picosecond carrier trapping dynamics, trapped carrier induced absorption without saturation at 10$^{20}$ cm$^{-3}$ carrier density, and polarized trap emission in single crystal. Similar spectroscopic characteristics have been observed in seemingly disparate materials where self-trapped carriers or excitons have been commonly observed[31].

Self-trapped carriers also differ strikingly from free carriers in transport properties. Because of strong localization to a single site, the former moves incoherently through thermally activated hopping process which moves faster with temperature while the latter moves through coherent band-like transport with inversely temperature relationship[30,31]. Temperature-dependent electrical measurements have been performed on Sb$_2$S$_3$ polycrystalline films and single crystals[35–37]. A combined electric and magnetic measurement on Sb$_2$S$_3$ crystal show thermal-assisted hopping transport mechanism[35], which can be well explained by carrier self-trapping in Sb$_2$S$_3$.

For free carriers, hole self-trapping instead of electron is generally observed in metal halides (e.g., AgCl, NaCl) and

chalcogenides (e.g., As$_2$S$_3$, As$_2$Se$_3$) because of rich $p$ orbital electrons in valance band, possible bond alternation, and small valance band width[31,32,38–40]. In additional to sulfide $p$ electrons, Sb$_2$S$_3$ possess Sb 5 $s^2$ inert lone-pair electrons with complex Sb–S chemical bonds and coordinations[2,41]. Electronic structure calculation also shows narrower valance band width than conduction band in Sb$_2$S$_3$ (Supplementary Fig. 1). All these suggest hole is more likely to be self-trapped than electron in Sb$_2$S$_3$. This is consistent with smaller hole mobility[21] than electron[17] and previously suggested accepter-like trap state in Sb$_2$S$_3$ crystal[35].

Self-trapping of electrons and holes after photoexcitation eventually leads to self-trapped excitons (STEs)[31]. Based on Stokes shift, the trapping depth of STE is ~0.6 eV away from band-edge state. A temperature dependent study (between 77 K and 297 K) on STE emission from Sb$_2$S$_3$ single crystal shows negligible band-edge exciton emission and no change of STE emission intensity as a function temperature, which indicates small energy barrier (less than kT of ~6.6 meV for 77 K) from band-edge carrier to STE and large (>0.4 eV) trapping depth for STE based on a thermal quenching model simulation (see

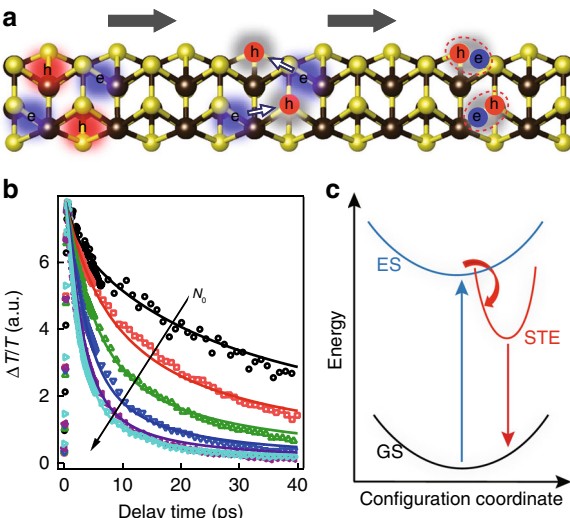

**Fig. 4 Self-trapping process. a** Scheme showing two-step formation process of STEs in $Sb_2S_3$: hole is self-trapped first and then electron is captured by trapped hole to form STE. **b** Carrier-trapping kinetics (open symbols) under different photoexcited carrier densities $N_0$ ($5 \times 10^{18}$ cm$^{-3}$ to $1.2 \times 10^{20}$ cm$^{-3}$) for $Sb_2S_3$ single-crystal flake and their fits to two-step formation mechanism (solid lines). **c** Qualitative adiabatic potential energy curve showing photoexcitation from ground state (GS) to excited state (ES), which further evolves into STE state by deforming lattice and losing substantial energy

Supplementary Fig. 7 and Supplementary Note 3). STE can form through either direct exciton trapping of same photoexcited electron–hole pair or through a two-step mechanism: hole self-trapping occurs first through mono-molecular process and then electron is captured by self-trapped hole (STH) to form STE through bimolecular process (Fig. 4a). The underlying physics in two-step mechanism is that photoexcitation generates free electrons and holes and electrons can move away from their geminate holes and sample a certain volume before being captured by STH. These two mechanisms can be differentiated by examining the trapping process as a function of photoexcited carrier density[32]. Trapping process does not depend on density in first case, but increase in second one as electron-STH bimolecular trapping would be proportional to the number of pre-formed STH and electron. As shown in Fig. 4b for $Sb_2S_3$ single-crystal flake, free carrier-trapping kinetics (obtained from visible TA measurement) decay faster with increasing photoexcited carrier density (same density value as in Fig. 3f) and approaches saturation. The density-dependent decay kinetics can be well replicated by a two-step rate equation model with an intrinsic hole-trapping rate constant ($k_{HT}$) and a bimolecular electron-STH capturing rate constant (Supplementary Note 4). We note in this model, we assume these two rate constants are independent on photo-excitation density in the investigated regime. The decay kinetics at high carrier density is limited by initial hole self-trapping step, thus approaches saturation. Fitting to the experimental kinetics (Fig. 4b) yields a hole intrinsic self-trapping lifetime ($1/k_{HT}$) of 1.8 ps and unambiguously demonstrates two-step trapping process for STE formation in $Sb_2S_3$ (Fig. 4a). In view of the large energy dissipated, it is likely bond alternation is involved for STE formation in $Sb_2S_3$. A combined study of advanced spectroscopic techniques (e.g., transient extreme-ultraviolet[42] or X-ray absorption[43]) and theoretical calculations is required to identify local chemical and structural information of STE.

STE in the $Sb_2S_3$ polycrystalline film has a half-life time of 23 ns (Supplementary Fig. 8), persisting through the decay of TA

signal. With reported photoexcitation diffusion coefficient of $6.8 \times 10^{-2}$ cm$^2$ s$^{-1}$ in the $Sb_2S_3$ polycrystalline film[21], this corresponds to a diffusion length of 400 nm. We note this diffusion coefficient value is likely a mixture of free carrier and STE[23] therefore this diffusion length should be considered as an upper bound for STE. This explains near-unity carrier collection efficiency in a $Sb_2S_3$ thin film solar cell, despite the formation of STE[8,10]. Since self-trapping is intrinsic to the crystal, equivalent sites occur in each unit cell, which allows thermal-assisted hopping transport. Indeed, the STE half-life time is significant shortened to be 6.9 ns on CdS/FTO substrate, suggesting efficient carrier extraction by transport in bulk followed by interfacial electron transfer to CdS (Supplementary Fig. 8). We also note there is band bending and build-in electric field in CdS/$Sb_2S_3$ junction, which can facilitate the dissociation of STE and carrier drift transport[10,14].

Similar to extrinsic defect states, self-trapping localizes carriers to specific sites and creates defect states in the gap (Fig. 4c). While in general extrinsic defects undeniably affect excited state carrier lifetime and solar cell performance, self-trapping is primarily responsible for ultrafast carrier trapping and energy loss process in $Sb_2S_3$, instead of surface/interface/bulk extrinsic trap states usually assumed. Based on PL Stokes shift, self-trapping causes 0.5–0.6 eV energy loss, which well explains the near-clamped $V_{oc}$ loss (0.63 V) between record $V_{oc}$ value and theoretical value. Different from extrinsic trap states, self-trapping will exist even in perfect $Sb_2S_3$ crystal, setting the upper limit on $V_{oc}$ and PCE of $Sb_2S_3$ solar cell. Assuming thermodynamic limit values of $J_{sc}$ and FF remain same in $Sb_2S_3$, but $V_{oc}$ is reduced from 1.4 V to 0.8 V due to self-trapping energy loss, maximum PCE of $Sb_2S_3$ solar cell will be 16% instead of 28.6%[2]. Although our studies here are all based on planer $Sb_2S_3$ thin films, carrier/exciton self-trapping should also occur in $Sb_2S_3$-sensitized photovoltaic devices with several nanometer thickness because of its intrinsic nature. There both extrinsic surface trapping and intrinsic self-trapping would affect excited-state carrier properties and device performances due to large surface area[5]. This study here calls for reconsideration of $Sb_2S_3$ and $Sb_2Se_3$ as the photovoltaic materials and designing and optimizing their optoelectronic devices.

## Methods

**Sample preparation.** The $Sb_2S_3$ polycrystalline film on the CdS/FTO substrate was grown via hydrothermal method following our previous report[10]. $Sb_2S_3$ poly-crystalline film on glass was prepared with spin-coating antimony-complex pre-cursor solution and post annealing[13]. Thermal-evaporated $Sb_2S_3$ thin film was deposited on glass at a high vacuum ($\sim 10^{-4}$ Pa) and annealed on preheated hot plate at 300 °C for 2 min in a glove box. A $Sb_2S_3$ single crystal was grown by chemical vapor transport (Shanghai Onway Technology Co, Ltd)[28].

**Device fabrication and measurement.** The hole-transporting layer was prepared by spin-coating Spiro-OMeTAD chlorobenzene solution with the concentration of 36 mg mL$^{-1}$ at 3000 rpm for 30 s and then with a post treatment at 100 °C for 10 min. Au counter electrode about 70 nm was deposited by a thermal evaporator under a pressure of $5.0 \times 10^{-4}$ Pa. Current−voltage measurements of $Sb_2S_3$ solar cell was performed in a standard xenon-lamp-based solar simulator (Oriel Sol 3A, USA). The test was under a 100 mW cm$^{-2}$ solar irradiation at room temperature. The solar simulator illumination intensity was calibrated by a monocrystalline silicon reference cell calibrated by the National Renewable Energy Laboratory (NREL). The external quantum efficiency (EQE, Model SPIEQ200) was measured using a single-source illumination system (halogen lamp) combined with a monochromator.

**Electronic structure calculation.** The calculations were carried out using Vienna ab initio simulation package (VASP) with a plane wave basis set. The electron–nuclei interaction was described by the projector augmented wave (PAW) method. For the exchange-correlation functional, we used the generalized gradient approximation of Perdew–Burke–Ernzerhof (GGA-PBE). Structures were fully relaxed until residual forces on constituent atoms become smaller than 0.01 eV Å$^{-1}$, and total electronic energies were converged to $10^{-5}$ eV. An energy cutoff parameter of 450 eV and a Monkhorst-Pack k-point sampling grid of $5 \times 5 \times$

15 for unit cell and $3 \times 3 \times 8$ for supercell were sufficient for convergence. We took into account the van der Waals interaction using a DFT-D2 approach (a nonlocal correction functional was added to account for dispersion interactions).

**Optical measurement**. Absorption spectra of films were measured on a Cary 5000 UV-Vis-NIR absorption spectrometer with integrating sphere. Absorption of exfoliated flakes and room temperature photoluminescence (PL) were performed on a home-built microscope setup. We used a 532 -nm CW laser as excitation and collected the PL and sent to an EMCCD (ProEM: 16002, Princeton Instrument) for visible region and a liquid nitrogen cooled InGaAs detector (PyLon IR, Princeton Instrument) for near-IR region. For femtosecond TA measurements, the fundamental beam from Yb: KGW laser (Pharos, Light Conversion Ltd.) was separated to multiple paths and sent to ultrafast spectrometer (TA100, Time-Tech Spectra, LLC). One was introduced into a noncollinear optical parametric amplifier to generate pump pulse at a certain wavelength. Another path was focused onto a YAG crystal to produce white light continuum (520–950 nm) as probe light. The third one was introduced to a collinear optical parametric amplifier for mid-IR probe generation. Nanosecond TA measurements were carried out on ns spectrometer (ns-TA100, Time-Tech Spectra, LLC) with a white light supercontinuum laser as probe pulse. The transmitted probe light with ($T_{pump}$) and without ($T_{unpump}$) pump were collected and the normalized transmittance change $\Delta T/T$ was calculated by $\Delta T/T = (T_{pump} - T_{unpump})/T_{unpump}$.

## Data availability

The source data necessary to support the findings of this paper are available from the corresponding author upon request.

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

## Acknowledgements

We thank the financial support from the National Natural Science Foundation of China (21773208, U1732150, and 11620101003) and National Key Research and Development Program of China (2017YFA0207700, 2016YFA0200604, and 2017YFA0204904).

## Author contributions

H.Z. conceived the idea and designed the experiments. X.W. and T.C. prepared all the polycrystalline film samples and characterized solar cell devices. Z.Y. and Y.C. performed all spectroscopic studies and data analysis. Z.Z. and J.Z. carried out electronic structure

calculations. Z.C. and Y.Y. performed electric characterization on single crystal. W.X. and W.L. helped with nanosecond TA measurements. Z.Y., X.W., and H.Z. wrote the paper together, and all authors comment on the paper.

## Competing interests

The authors declare no competing interests.
