## [Peer Review File · Nature Communications]

Reviewers' comments:

Reviewer #1 (Remarks to the Author):

In this manuscript, the ultrafast dynamics of free and trapped photocarriers in Sb₂S₃ have been studied by using multiple spectroscopic techniques. The major finding of this study reveals the evidences of self-trapped excitons, which obviously is of great interest to both semiconductor and photovoltaic communities. The discussion is insightful. The spectroscopic results are well presented. Overall, the manuscript is well-written, and also show a high level of novelty.

However, following concerns should be addressed before publication.

Major:

1. What are the binding energies of STE and hole polaron?
2. In figure 4b, are the kinetic traces obtained from vis-TA or Mid-IR TA? Since the mid-IR kinetics should unambiguously probe free carrier dynamics, it is the best that using mid-IR kinetic traces to study the STE formation dynamics. If the kinetics decay is governed by bi-molecular rate equation, then the inverse of TA signal should be proportional to the delay time. Such plot can be used as a strong evidence to support the proposed decay mechanism.
3. As the excitation intensity increases, the screening effect will start to exert influence on the STE formation. The binding energy should be decreased, and the yield of STE formation should be compromised. Evidences should be provided to demonstrate that the quantum yield of STE formation from free carriers keeps constant as the excitation intensity increases. If such effects are not taken into consideration, the model of the kinetics fit for Fig.4b could be invalid, and the determination of STH formation rate as well as STE formation rate could be inaccurate.
4. As the excitation intensity increases, the mean distance between trapped hole and electron decreases. Thus, the STE formation can be accelerated. For the same reason, this effect should also be taken into account in the kinetics fitting model (Fig. 4b).
5. The diffusion coefficient of photocarriers is mixture of free carriers and STE. To extract the STE diffusion coefficient, the pure diffusion coefficient of free carriers should be known first. Because the free carrier diffusion coefficient is much larger than that of STE, STE diffusion length should be shorter than ~400nm.
6. When the Sb₂Se₃ deposited on CdS/FTO, there is depletion region formed at the interface, where a built-in field exists. Under an electric field, the dissociation of STEs should be enhanced, and the transport is dominated by drifting rather than diffusion, which should be noted in the manuscript.
7. The PL emission arises from STE. However, the STE should be dark. Therefore, phonons should be involved in the process of photoluminescence, and the Stokes shift should also take into account the phonon energy in addition to the polaron binding energy and STE binding energy.

Minor:

1. The assignment of mid-IR absorption in this manuscript is not quite clear. If the Drude model is exploited to simulate the spectral shape, why the power is 2? Instead, the mid-IR absorption could also be caused by the phonon assisted intraband optical transition. In this model, the power is governed by the type of phonon species.
2. What are the lifetimes of free carriers extracted from Fig. 2 and Fig. 3?
3. Is that possible to estimate the Coulomb capture radius of the trapped hole?
4. Will the free carrier decay kinetics vary with excitation intensity for polycrystalline samples?

Reviewer #2 (Remarks to the Author):

In this submission, Z. L. Yang et. al. performed optical spectroscopy study on Sb₂S₃ polycrystalline film and single crystals and observed commonly existed red-shifted photoluminescence with large Stoke shift, picosecond free carrier trapping process and no obvious saturation even at excitation intensity as high as 1E20 cm⁻³. A further strong evidence of polarized emission make the authors ascribed these observations to the self-trapped excitons by this soft, quasi-1D Sb₂S₃ via first hole trapping followed by the electron trapping. Judged from the Stoke shift, they further estimated a Voc upper limit of 0.8 V and a theoretical efficiency limit of 16%, which cast doubt on the promise of antimony chalcogenide photovoltaics. Overall I think this is an in-depth study on Sb₂S₃ materials and provide a very new and convincing explanation on why the clamped low Voc of Sb₂S₃ photovoltaics, which would be appreciated by researchers in this field, and could also guide the understanding of more broadly solar cells employing semiconductors with low-dimensional crystal structures. I thus suggest acceptance of this manuscript upon minor revisions:

1. The authors said that "the TA data of Sb₂S₃ polycrystalline films can be well described by two principle components (A) and B as in figure 2b. However, they didn't provide any explanation to justify their conclusion. I see no correlation between data in Figure 2a and 2b. Also, the authors also analyzed their data by SVD method; however the necessary background introduction about this SVD technique should be provided.
2. In Figure 2d, the y-axis is $\Delta T/T$; please explain the physical meaning of it.
3. The authors provide single crystalline Sb₂S₃ as a strong evidence to support their STE understanding. However, the quality of single crystalline is not confirmed at all. Likely this is a poor crystal with a huge amount of defects. Please provide rocking curve or other characterization results to confirm its high quality.
4. Despite the authors provides Sb₂S₃ thin films (produced by two solution methods) and single crystals, I would still recommend the authors carry out similar study on thermally evaporated Sb₂S₃ thin films to check this is still the case. The authors want to draw an universal conclusion so that more solid data should be provided.
5. The authors mentioned the soft lattice and the elastic constants of Sb₂S₃; please direct provide the number of elastic constant. Furthermore, the Huang-Phys factor is generally used to quantitatively determine the interaction between phonon and exciton. Please measure this factor and compare with other known compound like NaCl or CsPbBr₃.
6. The STE emission nature could be further confirmed by the temperature dependent photoluminescence measurement. If STE is the case, at low temperature free exciton emission should be observed with simultaneously weakened STE emission. Please provide such data to confirm.
7. In the single crystal study, how about the effect of surface? Is it possible the observed STE emission and dynamics just due to carrier bulk diffusion to the surface defects/traps?
8. Any comment on how to solve the STE problem? It seems to be the fundamental problem that limits the efficiency limit of antimony chalcogenide photovoltaics.

Reviewer #3 (Remarks to the Author):

The authors deeply studied defects affecting Sb₂S₃ device performance based on investigation of three different samples, i.e. hydrothermally-synthesized sample, spin-coated sample, and single

crystal. Through the study, they found that ultrafast carrier trapping in Sb₂S₃ is mainly associated with intrinsic trapping rather than extrinsic defects. These phenomena were similarly observed from three different samples. This is an interesting study and gives some clues for understanding why the Sb₂S₃ has a low Voc. However, this conclusion rises some questions because of some unsolved issues. Therefore, the authors must address some issues to be published in this high-quality journal.

1. I am wondering if their conclusion is reasonable because the defect formation can be highly varied depending on variable factors, such as fabrication method/condition and device architecture. Are there any differences depending on the fabrication conditions, such as film thickness and Sb/S ratio? Is it impossible to distinguish the differences with your spectroscopic techniques? If there are some differences depending on fabrication conditions or some technical limitation, the authors must clearly mention these points.

2. I think that your conclusion may be fitted for the planar Sb₂S₃ film having hundreds of nanometer thickness. Do you think that your conclusion can be applicable to Sb₂S₃-sensitized device, where Sb₂S₃ has a thickness of several nanometers? The extrinsic defects would be dominant in case of Sb₂S₃-sensitized device because of their very small scale (please see ref. 5; Chem Soc. Rev. 2018, 47, 7659). If my opinion is right, your work is not common and may be suitable for any particular case. Therefore, you should seriously reconsider the use of terms, such as general and rationalized, in your work, especially in Abstract.

3. I found a minor error about the use of abbreviation. The abbreviation SVD for singular value decomposition is described three times in the manuscript (pages 3, 10, and 12). It should be mentioned only once in the manuscript.

Reviewer #1 (Remarks to the Author):

In this manuscript, the ultrafast dynamics of free and trapped photocarriers in Sb₂S₃ have been studied by using multiple spectroscopic techniques. The major finding of this study reveals the evidences of self-trapped excitons, which obviously is of great interest to both semiconductor and photovoltaic communities. The discussion is insightful. The spectroscopic results are well presented. Overall, the manuscript is well-written, and also show a high level of novelty.

Responses: We thank reviewer for the positive comments.

However, following concerns should be addressed before publication.

Major:

1. What are the binding energies of STE and hole polaron?

Responses: For binding energy, we believe reviewer refers to the ionization energy from localized trap state to free carrier state.

1) Based on energy difference between free carrier bandgap and PL from STE (i.e. Stokes shift of STE PL), the STE binding energy would be ~ 0.6 eV. To further confirm this, in the revised manuscript, we performed a temperature dependent PL measurement and we observed negligible intensity change of STE emission in temperature range 77 K \sim 297 K. A thermal quenching simulation indicates the ionization energy is larger than 0.4 eV which is consistent with 0.6 eV Stokes shift.

2) Optical excitation creates both electron and hole simultaneously therefore we cannot provide experimental binding energy value of hole polaron itself. Based on the discussion in main text that STE is formed due to hole self-trapping and electron-self trapped hole interaction should be weak based on PL intensity and STE lifetime, we would say the binding energy of hole polaron should be similar to that of STE.

Revisions:

In SI, we added content and figure about temperature dependent PL intensity of Sb₂S₃ single crystal flake and thermal quenching modeling

In main text, page 17, top, we added

...trapping of electrons and holes after photoexcitation generates self-trapped excitons (STEs).³⁰ Based on Stokes shift, the trapping depth of STE from band edge exciton state is about 0.6 eV. A temperature dependent study (77K ~ 297K) on STE emission from Sb_2S_3 single crystal shows negligible band edge exciton emission and no change of STE emission intensity as function temperature (Fig. S6), which indicates small barrier ($< kT \sim 6.6$ meV for 77 K) from band edge carrier to STE but large (> 0.4 eV) trapping depth for STE based on a thermal quenching model simulation (S7). STE can form through...

2. In figure 4b, are the kinetic traces obtained from vis-TA or Mid-IR TA? Since the mid-IR kinetics should unambiguously probe free carrier dynamics, it is the best that using mid-IR kinetic traces to study the STE formation dynamics. If the kinetics decay is governed by bi-molecular rate equation, then the inverse of TA signal should be proportional to the delay time. Such plot can be used as a strong evidence to support the proposed decay mechanism.

Responses:

1) We sorry for missing information. It's from vis-TA. As we have shown by comparison in Fig. 2 and Fig. 3, both visible TA and mid-IR yield the free carrier population kinetics and they two agree very well. Because of much better signal-to noise ratio for visible TA, in power dependent study, we rely on visible TA kinetics.

2) Thanks for the suggestion. In fact, the free carrier decay process or STE formation process is not simply a bi-molecular process. As discussed in manuscript, it's a two-step process: the first step is a monomolecular hole self-trapping process and the second step is a bimolecular process between electron and self-trapped hole. Therefore, the suggestion "plot of inverse of TA signal should be proportional to the delay time" will not apply here. (This only applies to bi-molecular kinetics) We have a

detailed kinetic model to analyze the kinetics as in main text and SI5.

Revisions: in main text, page 16, we added

...free carrier trapping kinetics (*obtained from visible TA measurement*) decay faster...

3. As the excitation intensity increases, the screening effect will start to exert influence on the STE formation. The binding energy should be decreased, and the yield of STE formation should be compromised. Evidences should be provided to demonstrate that the quantum yield of STE formation from free carriers keeps constant as the excitation intensity increases. If such effects are not taken into consideration, the model of the kinetics fit for Fig.4b could be invalid, and the determination of STH formation rate as well as STE formation rate could be inaccurate.

Responses: We thank reviewer raising this interesting point.

1) We believe fundamentally it's due to the quasi-particle size compared to lattice size. Indeed, for band edge carrier or large polaron, it's well known that with increasing carrier density, the inter-carrier/large polaron distance approaches carrier/large polaron size thus the screening induced by photoexcited carrier/large polaron will become prominent, leading to e.g. plasma regime. For small polaron, because of its tight localization within unit cell (usually to a single chemical bond), the carrier density required for small polaron to feel each other would be extremely high ($> 10^{23} \text{ cm}^{-3}$ based on unit cell size). Also, the small polaron is a quasi-particle with carrier dressed by polarized local lattice. Therefore, once small polaron starts to form due to carrier-lattice interaction, the lattice deformation will quickly screen the carrier-carrier interaction. Simply speaking, the polarizable lattice density is much larger than photoexcited carrier and the scattering and interaction between carrier-lattice is much stronger, which will further weaken the carrier-carrier interaction during self-trapping. Therefore, in our model, we can assume carrier density independent hole self-trapping process in this photoexcitation regime.

2) The linear increase of STE signal with photoexcitation is shown in Fig 2d and 3f, which indicates constant STE formation yield.

Revisions: in main text, page 16, we added

... (see SI5 for details). *We note in this model, we assume these two rate constants are independent on photoexcitation density in this regime. The ...*

4. As the excitation intensity increases, the mean distance between trapped hole and electron decreases. Thus, the STE formation can be accelerated. For the same reason, this effect should also be taken into account in the kinetics fitting model (Fig. 4b).

Responses: this kinetics model in manuscript has already considered the bimolecular collision between trapped hole and electron (the second step in the kinetic model). That's why the kinetics decays faster with increasing photoexcitation density. In the bi-molecular model, the distance is already reflected on density, which requires no additional distance factor.

5. The diffusion coefficient of photocarriers is mixture of free carriers and STE. To extract the STE diffusion coefficient, the pure diffusion coefficient of free carriers should be known first. Because the free carrier diffusion coefficient is much larger than that of STE, STE diffusion length should be shorter than ~400nm.

Responses: Thanks reviewer pointing this out. We simply took the diffusion co-efficient value from literature and didn't consider the issue of mixture of both free carrier and STE. In this regard, the real diffusion coefficient of STE and the diffusion length should be less. We made revisions accordingly.

Revisions: in main text, page 17, we added and cited necessary reference

...~ 400 nm. *We note this diffusion coefficient value is likely a mixture of free carrier and STE therefore this diffusion length should be considered as an upper bound. This explains near...*

6. When the Sb₂S₃ deposited on CdS/FTO, there is depletion region formed at the interface, where a built-in field exists. Under an electric field, the dissociation of STEs should be enhanced, and the transport is dominated by drifting rather than diffusion, which should be noted in the manuscript.

Responses: We thank reviewer pointing out this missing point. We added this point in the revised manuscript.

Revisions: in main text, page 18, we added

...electron transfer to CdS (Fig. SI4). *We note there is band bending and build-in electric field in CdS/Sb₂S₃ junction, which can facilitate the dissociation of STE and carrier drift transport.*

7. The PL emission arises from STE. However, the STE should be dark. Therefore, phonons should be involved in the process of photoluminescence, and the Stokes shift should also take into account the phonon energy in addition to the polaron binding energy and STE binding energy.

Responses:

1) It's not clear to me why STE should be dark. Electron and hole are spatially localized together therefore they have spatial overlap, although that's small. Because of tight localization, momentum is no longer a good quantum number thus no momentum-indirect issue. The only thing left is spin. Due to exchange interaction, triplet like STE is likely lower in energy than singlet like STE. But because of small spatial overlap, the exchange splitting should be very small. Therefore STE can still emit without phonon participation. The new added temperature independent PL intensity indicates this.

2) Bright PL from STE with high PL QY is actually very common, e.g. in metal halides or halide perovskite. see ref *Nature* 563, 541–545 (2018)

Revisions: In SI, we added content and Figure about temperature dependent PL intensity of Sb₂S₃ single crystal flake and thermal quenching modeling

Minor:

1. The assignment of mid-IR absorption in this manuscript is not quite clear. If the Drude model is exploited to simulate the spectral shape, why the power is 2? Instead, the mid-IR absorption could also be caused by the phonon assisted intraband optical transition. In this model, the power is governed by the type of phonon species.

Responses:

1) for free carrier absorption in semiconductor band, the absorption coefficient in Drude model is inversely proportional to ω^2 , thus proportional to λ^2 . (see ref. *IEEE Journal of Solid-State Circuits* 1978, 13 (1), 180-187; *IEEE Journal of Quantum Electronics* 1990, 26 (1), 113-122)

$$\alpha = \frac{q^3 \lambda^2 p}{4 \pi^2 \epsilon_0 c^3 n m^{*2} \mu} \quad (1)$$

where λ = wavelength, p = density of free carriers (either electrons or holes), n = refractive index, m^* = effective mass, and μ = mobility. Substitution of numerical values

2) Free carrier absorption in degenerate semiconductor conduction/valance band,

fundamentally, is an intraband absorption, generally as $\alpha_{\text{FC}} \propto \lambda^n$. Both $n = 2, 2.5, 3$ has been reported, depending detailed scattering mechanism and wavelength region. In the mostly commonly used Drude model treatment, it is proportional to λ^2 at long wavelength, which has been confirmed by many experimental results (See reference cited above). In our analysis here, we do not try to differentiate different scattering mechanism but just show mid-IR is contributed by free carrier absorption.

2. What are the lifetimes of free carriers extracted from Fig. 2 and Fig. 3?

Responses: because free carrier decay process is not a mono-molecular process but a two-step process with second step through bi-molecular process, we cannot define a simple “lifetime” (lifetime now depends on specific photoexcitation density). Instead we have relied on kinetic mode to extract the rate constant for STH and STE formation. Because STH is still a mono-molecular process, a lifetime for hole can be defined ~ 1.8 ps as already discussed in manuscript.

3. Is that possible to estimate the Coulomb capture radius of the trapped hole?

Responses: a simple and general way to estimate the Coulomb capture radius is by letting Coulombic interaction energy equal to thermal energy

$$\frac{e^2}{4\pi\epsilon r} = \frac{3}{2}kT$$

with DC dielectric constant of ~ 7 for Sb_2S_3 . This corresponds to a capture radius of ~ 5 nm for free carrier. For trapped hole, dielectric constant surrounding that should be much larger due to lattice deformation. Thus the capture radius should be much smaller than 5 nm. Without detailed atomic information, we cannot give accurate estimate at present.

4. Will the free carrier decay kinetics vary with excitation intensity for polycrystalline samples?

Responses: Similar to single crystal sample, we also observe faster free carrier decay process with increasing excitation density, indicating two-step mechanism as described in main text. In manuscript, we focus on single crystal result to extract intrinsic material properties.

Reviewer #2 (Remarks to the Author):

In this submission, Z. L. Yang et. al. performed optical spectroscopy study on Sb₂S₃ polycrystalline film and single crystals and observed commonly existed red-shifted photoluminescence with large Stoke shift, picosecond free carrier trapping process and no obvious saturation even at excitation intensity as high as 1E20 cm⁻³. A further strong evidence of polarized emission make the authors ascribed these observations to the self-trapped excitons by this soft, quasi-1D Sb₂S₃ via first hole trapping followed by the electron trapping. Judged from the Stoke shift, they further estimated a Voc upper limit of 0.8 V and a theoretical efficiency limit of 16%, which cast doubt on the promise of antimony chalcogenide photovoltaics. Overall I think this is an in-depth study on Sb₂S₃ materials and provide a very new and convincing explanation on why the clamped low Voc of Sb₂S₃ photovoltaics, which would be appreciated by researchers in this field, and could also guide the understanding of more broadly solar cells employing semiconductors with low-dimensional crystal structures. I thus suggest acceptance of this manuscript upon minor revisions:

Responses: We thank reviewer for the positive comments.

1. The authors said that "the TA data of Sb₂S₃ polycrystalline films can be well described by two principle components (A) and B as in figure 2b. However, they didn't provide any explanation to justify their conclusion. I see no correlation between data in Figure 2a and 2b. Also, the authors also analyzed their data by SVD method; however the necessary background introduction about this SVD technique should be provided.

Responses: We are sorry that we didn't make it clear in the manuscript about this

analysis method. SVD is a general method for analyze complex TA spectra. In the revised manuscript, we added more introduction and discussion on that and cite a review article about that.

Revisions: on main text page 8, we added the following content and necessary reference

...We observed clear spectral evolution with increasing delay time, indicating the presence of multiple (at least two) transient absorbing species and the conversion between them after photoexcitation. TA result is a two-dimensional data as a function of delay time and probe wavelength. For TA results with spectral and temporal separated species, it is common to do analysis with sets of discrete time and wavelength. For a complex TA result, singular value decomposition (SVD) based on time-wavelength separability provides a facial and general method to describe TA result with minimum number of transient species (base spectra) on a completely model-free basis.²⁶ And the emergence and evolution of the species can be followed individually with time (base time traces). In order to disentangle...

2. In Figure 2d, the y-axis is $\Delta T/T$; please explain the physical meaning of it.

Responses: we do TA measurement on transmittance mode. So what we really measure is the transmittance change with and without pump. We explained this in method part.

Revisions: page 21, method part, we added the detailed description

The transmitted probe light with (T_{pump}) and without (T_{unpump}) pump were collected and the normalized transmittance change $\Delta T/T$ was calculated by $\Delta T/T = (T_{\text{pump}} - T_{\text{unpump}}) / T_{\text{unpump}}$.

3. The authors provide single crystalline Sb₂S₃ as a strong evidence to support their STE understanding. However, the quality of single crystalline is not confirmed at all. Likely this is a poor crystal with a huge amount of defects. Please provide rocking curve or other characterization results to confirm its high quality.

Responses: we thank reviewer pointing out this missing information, which is indeed very important. In the revised manuscript, we performed SCLC measurement on Sb₂S₃ single crystal to determine the trap density. The value of $6.8 \times 10^9 \text{ cm}^{-3}$ is even smaller than that of MAPbI₃ single crystal, indicating high quality. Therefore the carrier trapping under investigated photoexcitation density can only be due to

self-trapping instead of extrinsic defects

Revisions:

In SI, we added SCLC characterization of Sb_2S_3 single crystal and determined the trap density.

In main content, on page 11, we added

...consistent with its quasi-1D crystal structure. *We characterized the trap density of Sb_2S_3 single crystal by space charge-limited current method which shows a very low trap density of $6.8 \times 10^9 \text{ cm}^{-3}$ (supplementary Fig. S3).* To probe the sample with...

on page 13, we added

which is too large to be related to extrinsic defects for Sb_2S_3 single crystal *with a defect density of $6.8 \times 10^9 \text{ cm}^{-3}$.*

4. Despite the authors provides Sb_2S_3 thin films (produced by two solution methods) and single crystals, I would still recommend the authors carry out similar study on thermally evaporated Sb_2S_3 thin films to check this is still the case. The authors want to draw a universal conclusion so that more solid data should be provided.

Responses: Thanks for reviewer's nice suggestion. To make our conclusion general and solid, in the revised manuscript, we followed reviewer's suggestion and added results from thermal-evaporated Sb_2S_3 thin films.

Revisions: in SI, we added results about thermally evaporated Sb_2S_3 polycrystalline film. (Fig. S6)

in main content, page 13, we revised

...We performed similar steady state and transient optical measurements on spin-coated and *thermal-evaporated Sb_2S_3 polycrystalline film (supplementary Fig. S5 and S6)* ...

5. The authors mentioned the soft lattice and the elastic constants of Sb_2S_3 ; please direct provide the number of elastic constant. Furthermore, the Huang-Phys factor is generally used to quantitatively determine the interaction between phonon and exciton. Please measure this factor and compare with other known compound like NaCl or CsPbBr_3 .

Responses: we thank viewer pointing out these missing information and we added them in the revised manuscript

1) we provided the elastic constant value of Sb_2S_3 (40) from literature in the revised

manuscript.

2) we estimated the Huang-Phys factor of Sb_2S_3 (~ 38.5) and compared it to NaCl, CsPbBr_3 and $\text{Cs}_2\text{AgInCl}_6$ these common compounds.

Revisions:

In SI, we added note about estimating Huang-Rhys parameter based on Stokes shift

In main content, on page 14, we added

The elastic constant of Sb_2S_3 have been calculated³² and the value (~ 40) is as small as that of SiO_2 and NaCl crystals.³¹ *The Huang-Rhys parameter of Sb_2S_3 , which reflects carrier-phonon interaction, was estimated to be ~ 38.5 , which as large as that of NaCl and $\text{Cs}_2\text{AgInCl}_6$ ³³ where STE has been demonstrated (S8). Dimensionality also plays an important...*

6. The STE emission nature could be further confirmed by the temperature dependent photoluminescence measurement. If STE is the case, at low temperature free exciton emission should be observed with simultaneously weakened STE emission. Please provide such data to confirm.

Responses: we thank reviewer raising this important point! We indeed performed a temperature dependent (77K - 297K) study on PL. We observed negligible band edge exciton emission and no change of STE emission intensity as function temperature. We also performed a thermal quenching simulation. This indicates small barrier ($< kT \sim 6.6$ meV for 77 K) from band edge carrier to STE but large (> 0.4 eV) trapping depth for STE. Considering such small activation barrier for STE formation, in order to observe what reviewer suggested, we need to do go to even lower temperature, which is not allowed in our group. Anyway, this temperature dependent already provide valuable information.

Revisions: In SI, we added content and figure about temperature dependent PL intensity of Sb_2S_3 single crystal flake and thermal quenching modeling

In main text, page 17, top, we added

...trapping of electrons and holes after photoexcitation generates self-trapped excitons (STEs).³⁰ *Based on Stokes shift, the trapping depth of STE from band edge exciton state is about 0.6 eV. A temperature dependent study (77K ~ 297K) on STE emission from Sb₂S₃ single crystal shows negligible band edge exciton emission and no change of STE emission intensity as function temperature (Fig. S6), which indicates small barrier ($< kT \sim 6.6$ meV for 77 K) from band edge carrier to STE but large (> 0.4 eV) trapping depth for STE based on a thermal quenching model simulation (S7). STE can form through...*

7. In the single crystal study, how about the effect of surface? Is it possible the observed STE emission and dynamics just due to carrier bulk diffusion to the surface defects/traps?

Responses:

1) for our single crystal study, we always exfoliated the single crystal to expose fresh surface. The layered crystal structure of Sb₂S₃ allows easy exfoliation with shape and clean surface without bond breaking. So the surface of single crystal flake is essentially same as the bulk in terms of crystal structure. This is different from conventional 3D semiconductor with dangling bond or vacancy at surface.

2) Based on the absorption coefficient, the light penetration depth is about 250 nm at our excitation wavelength. The hole lifetime in single crystal is estimated to be 1.8 ps. With free carrier diffusion coefficient ~ 10 cm² s⁻¹ (J. Phys. Chem. Lett. 2019, 10, 4881–4887), the diffusion length of free hole is less than 40 nm, much shorter than light penetration depth. Therefore what we have measured is the photoexcitation properties in bulk not on surface.

3) If it's due to surface effect, we would expect to see much faster decay process for polycrystalline sample than single crystal, which is not true in our experiment.

8. Any comment on how to solve the STE problem? It seems to be the fundamental problem that limits the efficiency limit of antimony chalcogenide photovoltaics.

Responses: As we have discussed in manuscript, STE formation is related to lattice softness, carrier-phonon interaction and dimensionality. All these are intrinsic properties of this material. Therefore, it seems to me STE pose a fundamental limitation of using this material for photovoltaic applications.

Reviewer #3 (Remarks to the Author):

The authors deeply studied defects affecting Sb₂S₃ device performance based on investigation of three different samples, i.e. hydrothermally-synthesized sample, spin-coated sample, and single crystal. Through the study, they found that ultrafast carrier trapping in Sb₂S₃ is mainly associated with intrinsic trapping rather than extrinsic defects. These phenomena were similarly observed from three different samples. This is an interesting study and gives some clues for understanding why the Sb₂S₃ has a low Voc. However, this conclusion rises some questions because of some unsolved issues. Therefore, the authors must address some issues to be published in this high-quality journal.

Responses: We thank reviewer for the positive comments and raised issues.

1. I am wondering if their conclusion is reasonable because the defect formation can be highly varied depending on variable factors, such as fabrication method/condition and device architecture. Are there any differences depending on the fabrication conditions, such as film thickness and Sb/S ratio? Is it impossible to distinguish the differences with your spectroscopic techniques? If there are some differences depending on fabrication conditions or some technical limitation, the authors must clearly mention these points.

Responses: we thank reviewer pointing out this question.

1) In revised manuscript, we added another thermal-evaporated Sb₂S₃ film for comparison in order to reach general and solid conclusion. We intent to study different samples from different growth methods. As reviewer pointed out, depending on the fabrication methods, the extrinsic defects (defect types and energetic levels) would be different for three polycrystalline and one single crystal samples. The key finding of our experiment is common spectroscopic and dynamics features in all four types of samples, in spite of different fabrication methods. Therefore, together with lattice properties, including elastic constant, Huang-Rhys factor and dimensionality argument, we reach the general conclusion about intrinsic self-trapping in Sb₂S₃ for low Voc and efficiency, instead of extrinsic defects.

2) The three deposition methods we use cover all commonly used methods in antimony chalcogenide photovoltaic community. For hydrothermal grown sample, we

also assembled solar cell device and characterized its performance. The power conversion efficiency and IPCE are all among top values in this field.

3) The thicknesses for investigated samples are all similar (~ 150 nm). The Sb/S for different deposition method could be different. The comparison of samples from different deposition method and single crystal is the key to illustrate our conclusion.

4) In the revised manuscript, we also characterized the trap density of our single crystal sample by SCLC method, which shows a value of $6.8 \times 10^9 \text{ cm}^{-3}$. This value is even slower than that of perovskite single crystal. But we still observe ultrafast carrier trapping process under much higher photoexcitation density, which makes us conclude carrier self-trapping instead of extrinsic trapping play the dominant role

Revisions:

In SI, we added new optical results for thermal evaporated Sb₂S₃ polycrystalline film (Fig. S6)

In SI, we added SCLC characterization on Sb₂S₃ single crystal, which shows very low trap density. (Fig. S3)

In main content, on page 11, we added

...consistent with its quasi-1D crystal structure. *We characterized the trap density of Sb₂S₃ single crystal by space charge-limited current method which shows a very low trap density of $6.8 \times 10^9 \text{ cm}^{-3}$ (supplementary Fig. S3).* To probe the sample with...

on page 13, we added

which is too large to be related to extrinsic defects for Sb₂S₃ single crystal *with a defect density of $6.8 \times 10^9 \text{ cm}^{-3}$.*

in main content, page 13, we revised

...We performed similar steady state and transient optical measurements on spin-coated and *thermal-evaporated Sb₂S₃ polycrystalline film (supplementary Fig. S5 and S6) ...*

2. I think that your conclusion may be fitted for the planar Sb₂S₃ film having hundreds of nanometer thickness. Do you think that your conclusion can be applicable to Sb₂S₃-sensitized device, where Sb₂S₃ has a thickness of several nanometers? The extrinsic defects would be dominant in case of Sb₂S₃-sensitized device because of their very small scale (please see ref. 5; Chem Soc. Rev. 2018, 47, 7659). If my opinion is right, your work is not common and may be suitable for any particular case. Therefore, you should seriously reconsider the use of terms, such as general and

rationalized, in your work, especially in Abstract.

Responses: We thank reviewer for raising this interesting point.

Based on our discussion in manuscript, the carrier/exciton self-trapping in Sb_2S_3 is related to the intrinsic properties of this material: lattice softness, carrier-phonon interaction and quasi-1D crystal structure. All these intrinsic properties should remain in both planer sample with hundreds of nanometer thickness and sensitized sample with several nanometers. Therefore the carrier self-trapping we concluded based on Sb_2S_3 polycrystalline films and single crystal should also occur in sensitized Sb_2S_3 with thickness of several manometer. Carrier self-trapping should be general for this material, regardless of thickness and preparation method.

As reviewer said, in sensitized structure with several nanometer thickness, the extrinsic defects will be severe compared to planer sample due to large surface area. In this case, both extrinsic and intrinsic trapping will both affect carrier lifetime and properties. Considering the fast hole trapping (1.8 ps) and deep trap depth (~ 0.6 eV below band gap) of self-trapping, the intrinsic self-trapping is like still play an important role.

Revisions: on page 19, we added the discussion about self-trapping in Sb_2S_3 sensitized devices

... will be 16% instead of 28.6%. *“Although our studies here are all based on planer Sb_2S_3 thin film samples, carrier/exciton self-trapping should also occur in Sb_2S_3 -sensitized samples with several nanometer thickness because of its intrinsic nature. There both extrinsic surface trapping and intrinsic self-trapping would affect excited state carrier properties and device performances due to large surface area.”*

This study here calls...

on page 1, abstract, we revised according to what reviewer suggested

also rationalizes the large V_{oc} loss and near-unity carrier collection efficiency in Sb_2S_3 thin film solar cell.

3. I found a minor error about the use of abbreviation. The abbreviation SVD for singular value decomposition is described three times in the manuscript (pages 3, 10, and 12). It should be mentioned only once in the manuscript.

Responses: We thank review pointing this error out. We made this correction and others in the revised manuscript.

Reviewers' comments:

Reviewer #1 (Remarks to the Author):

In the revised manuscript, there is a minor issue regarding the diffusion length. As the free carrier mobility is much larger than STE even though its lifetime is much shorter, it will make more sense to differentiate the STE diffusion length and free carrier diffusion length. Other than this, the authors have addressed all my concerns. I recommend this manuscript for publication.

Reviewer #2 (Remarks to the Author):

In this revision, the authors have made substantial efforts toward my questions. While most of them are solved now, there are still some questions need to be addressed:

1. Confirmation of high quality of the Sb₂Se₃ single crystals via the low trap density obtained from SCLC result is not acceptable. Providing of solid XRD evidence like rocking curve (is a must, which is easily accessible these days.

2. The authors concluded that the Huang-Phys constant as 38.5, however they didn't provide details how this number is obtained.

3. No PL intensity or shape is change during the temperature dependent PL measurement; the authors believe that the small barriers between FE and STE (~6.6 meV) is the reason. This is somewhat unusual in most materials a huge change could be detected. I would suggest the authors go to liquid He temperature to confirm this, as the conclusion drawn by the authors have a large impact on this field so we should 100% sure about the results. Thanks the authors for the efforts.

Reviewer #3 (Remarks to the Author):

The authors answered all my previous questions sincerely. Therefore, I think that current manuscript is acceptable for publication in Nat. Commun.

Reviewer #1 (Remarks to the Author):

In the revised manuscript, there is a minor issue regarding the diffusion length. As the free carrier mobility is much larger than STE even though its lifetime is much shorter, it will make more sense to differentiate the STE diffusion length and free carrier diffusion length. Other than this, the authors have addressed all my concerns. I recommend this manuscript for publication.

Response: We thank reviewer for the recommendation and raising this minor point. In this study, we only focus on establishing of STE and the carrier dynamics especially the STE formation process. Differentiate and determine the mobility and diffusion length of free carrier and STE is beyond the scope this study. We took the literature values of diffusion coefficient (which is mixture of both free carrier STE) and the lifetime of STE and estimated the diffusion length. As we have pointed out in manuscript, this value should be considered as an upper bound since the lifetime of free carrier is much shorter. The careful study on charge carrier transport process would be future work.

Reviewer #2 (Remarks to the Author):

In this revision, the authors have made substantial efforts toward my questions. While most of them are solved now, there are still some questions need to be addressed:

1. Confirmation of high quality of the Sb₂Se₃ single crystals via the low trap density obtained from SCLC result is not acceptable. Providing of solid XRD evidence like rocking curve (is a must, which is easily accessible these days.

Response: Thanks reviewer's suggestion. In the first-round revision, we estimated electronic defect densities based on SCLC method. Comparing the defect density with photoexcitation density is meaningful and important to establish STE. For the XRD rocking curve, it's a pity we don't have this kind of facility in our university. We searched and found an institute with this capability. We discussed with them and they tried. Unfortunately, our sample is less than 1mm in width and height. Therefore, we cannot get the rocking curve results which typically requires a sample size of 4 ~ 5 mm. Following reviewer's suggestions, in the revised version, we performed and

added X-Ray Laue photograph of our single crystal. Both Laue cone and high indexing rate show high crystal quality. We hope reviewer can agree that this X-Ray result, together with SCLC characterization on trap densities, confirms high crystal quality.

Revisions: in SI3, Fig. 3, we added Transmission X-Ray Laue photograph results together with SCLC characterizations.

in main text, page 11, we added

...structure. *Transmission X-Ray Laue photograph of Sb_2S_3 single crystal indicates high crystalline quality.* We...

2. The authors concluded that the Huang-Phys constant as 38.5, however they didn't provide details how this number is obtained.

Response: Maybe reviewer missed it. We have a whole section in SI (S7) estimating Huang-Phys factor based on the Stokes shifted PL and the phonon mode strongly coupled to photoexcitation.

3. No PL intensity or shape is change during the temperature dependent PL measurement; the authors believe that the small barriers between FE and STE (~ 6.6 meV) is the reason. This is somewhat unusual in most materials a huge change could be detected. I would suggest the authors go to liquid He temperature to confirm this, as the conclusion drawn by the authors have a large impact on this field so we should 100% sure about the results. Thanks the authors for the efforts.

Response: Based on liquid nitrogen measurement, we estimated the barrier between FE and STE is < 6.6 meV. To further quantify the barrier height, lower temperature is required **if there indeed is a barrier**. We thank reviewer's suggestion. First, we have to admit we cannot find such system with the liquid He temperature and

high-sensitive near-IR detection in our university. Unlike visible, it's rare to have near-IR PL measurement at liquid He temperature. Second, more importantly, according to polaron theory as in book ref. 31, page 22, there is a barrier for three-dimensional system but in one-dimensional system, there is actually no barrier for self-trapping (see Fig. 1.7 a below). Considering the one-dimensional crystal structure of Sb_2S_3 , it is very likely there is no barrier (or really really small) for STE formation. The barrier based on liquid nitrogen measurement is already estimated to be less than 6.6 meV, which is essentially negligible at room temperature. Overall, performing liquid He measurement with Near-IR detection is not accessible and also not that meaningful at current stage.

[Redacted]

Revisions: Main content, page 14, we added:

Compared to three dimensional system, free carrier in (quasi) one-dimensional (1D) system has been predicted to be intrinsically unstable and tends to be self-trapped *without barrier*.

Reviewer #3 (Remarks to the Author):

The authors answered all my previous questions sincerely. Therefore, I think that current manuscript is acceptable for publication in Nat. Commun.

Responses: We thank reviewer for the recommendation.

REVIEWERS' COMMENTS:

Reviewer #1 (Remarks to the Author):

The reviewer is satisfied with the response.

Reviewer #2 (Remarks to the Author):

All my previous concerns are addressed.